

# Serpulid-microbialitic bioconstructions from the upper Sarmatian (Middle Miocene) of the Central Paratethys Sea (NW Hungary) – witnesses of a microbial sea

Mathias Harzhauser[1,2], Oleg Mandic[1], and Werner E. Piller[2]

[1]Geologisch-Paläontologische Abteilung, Naturhistorisches Museum Wien, Burgring 7, 1014 Wien, Austria
[2]Institut für Erdwissenschaften, Bereich Geologie und Paläontologie, Universität Graz, Heinrichstraße 26, 8010 Graz, Austria

*Correspondence to*: Mathias Harzhauser (mathias.harzhauser@nhm-wien.ac.at)

**Abstract.** We present so far unknown stacked bowl-shaped bioherms reaching a size of 45 cm in diameter and 40 cm in height from weakly solidified peloidal sand from the upper Sarmatian of the Paratethys Sea. The bioherms were mostly embedded in

sediment and the 'stages' reflect a reaction on sediment accretion and sinking into the soft sediment. The bioherms are spirorbid-microclots-acicular cement boundstones with densely packed *Janua* tubes surrounded by microclots and acicular cement solidifying the bioherm. The surrounding sediment is a thrombolite made of peloids and polylobate particles (mesoclots) which are solidified synsedimentarily by micrite cement and dog-tooth cement in a later stage. The shape of the bioherms reflects a series of growth stages with an initial stage ('start-up' phase) followed by a more massive 'keep-up stage'

which grades into a structure with a collar-like outer rim and a central protrusion and finally by a termination of growth ('give-up stage'). The setting was a shallow subtidal environment with normal marine or hypersaline, oligotrophic conditions with an elevated alkalinity. The stacked bowl-shaped microbialites are a unique feature so far undescribed. Modern and Neogene microbialite occurrences are no direct analogues to the described structures but the marine examples, like at the Bahamas, Shark Bay and the Persian Gulf offer insight into the microbial composition and environmental parameters.

The microbialites and the surrounding sediment document a predominance of microbial activity in the shallow marine environments of the Paratethys Sea during the late Middle Miocene, which was characterized by a warm, arid climate.

## 1 Introduction

The Miocene Central Paratethys Sea was a warm temperate to subtropical sea covering large parts of central and south-eastern Europe (Rögl, 1998; Popov et al., 2004; Harzhauser and Piller, 2007). Normal marine conditions prevailed during most of the

Badenian (Langhian and early Serravallian) with a moderately diverse scleractinian coral reef fauna (Riegl and Piller, 2000; Perrin and Bosellini, 2012; Wiedl et al. 2013). Microbialites were of subordinate importance in these ecosystems except for a short phase around 13.8 Ma at the Langhian/Serravallian boundary. At that time the Paratethyan basins became largely isolated from the Proto-Mediterraean Sea and hypersaline conditions of the Badenian Salinity Crisis resulted in the formation of evaporites (de Leeuw et al., 2010; Bukowski, 2011; Śliwiński et al., 2012). Stromatolites formed in restricted lagoons



(Harzhauser et al., 2014). Around 12.7 Ma, a major extinction event, termed Badenian-Sarmatian Extinction Event (BSEE) (Harzhauser and Piller, 2007; Palcu et al., 2015), caused a final collapse of Paratethyan coral reef communities, accompanied by a dramatic decline of calcareous red algae diversity. The BSEE marked the onset of the Sarmatian, which was characterized by a peculiar endemic marine fauna (Harzhauser and Piller, 2007). During the early Sarmatian, bryostromatolites established in coastal-lagoonal settings pointing to eutrophication and polyhaline salinities (Piller and Harzhauser, 2023 and references

therein). Around 12 Ma, climate warming and aridification changed the environmental conditions of the Sarmatian Paratethys and oligotrophic, normal marine to hypersaline conditions prevailed (Latal et al., 2004; Kranner et al., 2021). Oolite shoals became widespread during the late Sarmatian in the Central Paratethys Sea (Piller and Harzhauser 2005). At that time, the bryostromatolites had vanished and became replaced by another type of microbialites lacking bryozoans. These upper Sarmatian bioconstructions are classified as foraminiferal *Nubecularia* boundstones, *Nubecularia*-coralline algal boundstones,

stromatolitic/thrombolitic boundstone and serpulid-nubeculariid-microbial boundstones (Piller and Harzhauser, under review). The herein described structure is, however, different from the described boundstones in representing stacked bowl-shaped microbialites with a high share of the serpulid *Janua*. Many modern marine microbialites occur in oligotrophic intertidal – to shallow subtidal environments, often affected by aridity and hypersalinity (Siqueiros-Beltrones, 2008; Johnson et al., 2012; Mobberley et al., 2015). Photosynthetic cyanobacteria and proteobacteria are the most important groups in these microbialites

(Mobberley et al. 2015) but a broad consortium of other microbes contribute to the microbial mats of living structures including other bacteria, archaea, and various heterotrophic eukaryotes (Khodadad and Foster, 2012; Edgcomb et al., 2014).

## 2 Geographical and geological setting

Sarmatian microbialites were detected in 2004 in the Piuspuszta gravel pit in NW-Hungary (Fig. 1). The pit is situated in the Eisenstadt-Sopron Basin, which is a triangular Neogene structure of about 20 km width (Harzhauser, 2022). To the north, the

basin is confined by the NE-SW trending Leitha Mountains, while the Rust Mountains separate it from the Danube Basin to the east. A crystalline ridge, covered by Lower Miocene gravel, extending from the Rosalia Mountains to Brennbergbánya, defines the basin's southern margin. This topographical barrier separates the Eisenstadt-Sopron Basin from the Oberpullendorf and Styrian Basins. Sedimentation started during the Early Miocene with fluvial gravel and ended during the Late Miocene with lacustrine deposits of Lake Pannon (Harzhauser, 2022).

The gravel pit is located about 400 m SSW of the border crossing point St. Margarethen at the Hungarian-Austrian border and 3 km NNW of Fertörákos in Hungary (Fig. 1). The outcrop is generally abandoned but occasionally gravel is exploited for local use. Within the pit, upper Sarmatian and lower Pannonian deposits are exposed along two levels. The lower level is situated in the southern part of the pit and exposes about 8 m of upper Sarmatian marine gravel of a Gilbert-type delta passing into well sorted fine to medium sand with *Sarmatimactra vitaliana* and *Sarmatigibbula podolicus* and large, platy sandstone

concretions (47° 45.146' N, 16° 37.118' E). In the upper level of the pit, an about 10-m-thick succession crops out (Fig. 2) along an about 35-m-long NNW-SSE trending section. The upper level represents the overlying succession of the gravel and





sand of the lower level, but both occurrences are separated by a fault, associated with the Kőhida-fault zone (Rosta, 1993). The investigated bioherms were detected in 2004 in the eastern part of the gravel pit (47° 45.196' N, 16° 37.122' E).

During the last years, the Piuspuszta gravel pit was frequently visited during geological excursions, but only very limited published data is available for the outcrop. Rosta (1993) described parts of the section and identified the lower part of the outcrop as part of a Sarmatian Glibert-type delta. A lateral equivalent of the Piuspuszta section is outcropping south of St. Margarethen in Austria at the 'Altes Zollhaus' section (Fig. 1) from where Harzhauser and Kowalke (2002) and Harzhauser et al. (2002) described Sarmatian and Pannonian mollusk faunas and Piller and Harzhauser (under review) serpulid-nubeculariid-microbial boundstones. Latal et al. (2004) provided stable isotope data on some of the mollusks from St. Margarethen.

## 2.1 The Piuspuszta section

The section starts with 3 m of well sorted marine gravel, frequently showing imbrication, grading in the top into a 40-cm-thick layer of whitish caliche (units 1 and 2 in Fig. 2) (47° 45.248 'N, 16° 37.143' E). Above a strong relief of up to 15 cm amplitude follows an about 1.3-m-thick fining upwards unit of weakly solidified old rose peloidal calcareous sand with scattered gravel and abundant mollusks (unit 3). An about 15-cm-thick horizon with thin limonitic roots occurs in the upper part of the unit, where discontinuous caliche layers become frequent. Above, follows a 1.7-m-thick unit of weakly solidified, greenish-grey, peloidal calcareous sand with fine to coarse siliciclastic sand and occasional gravel stringers (unit 4). Discontinuous caliche layers and a rich mollusk assemblage occur in unit 4. The described bioherms start close above the base of unit 4 (47° 45.227' N, 16° 37.162' E), which is overlain by a succession of 1.2 m of gravel (unit 5), 1.8 m of greenish-greyish peloidal sand with abundant caliche layers (unit 6) and 0.7 m of gravel (unit 7). Unit 8 represents a 0.9-m-thick bed of old rose peloidal calcareous sand with siliciclastic fine to middle sand and gravel and frequent caliche layers. The upper part of the Sarmatian succession is formed by 1.8 m fossiliferous mixed carbonate-siliciclastic fine to middle sand with a thick caliche layer in the top (unit 9). Above follows an about 1-m-thick unit of poorly sorted fine to coarse sand with gravel, layers of small oncoids and frequent shells of *Melanopsis pseudonarzolina* (unit 10) (47° 45.238' N, 16° 37.163' E).

## 2.2 Biostratigraphy

The occurrence of the bivalve Sarmatimactra vitaliana in units 1 to 9 is indicative for the upper Sarmatian *Sarmatimactra* Zone (Papp, 1954; Harzhauser and Piller, 2004a, 2004b; Piller and Harzhauser, 2005), which roughly spans an interval from 11.9 to 11.6 Ma (Harzhauser and Piller, 2004a). This part of the Sarmatian corresponds to the lower Bessarabian stage of the Eastern Paratethys (Popov et al., 2004). The presence of *Melanopsis pseudonarzolina* places unit 10 into the lower Pannonian (Harzhauser et al., 2002).

## 2.3 Paleoenvironment

The Piuspuszta section is a lateral equivalent of the 'Altes Zollhaus' section at St. Margarethen in Austria, described by Harzhauser and Kowalke (2002) and Piller and Harzhauser (under review). The 'Altes Zollhaus' section is situated 1.2 km



NNE of the Piuspuszta section and differs mainly by its thicker limestone beds. Terrestrial and freshwater species and the oligohaline *Potamides hartbergensis* assemblage, which were described from the 'Altes Zollhaus' section by Harzhauser and

Kowalke (2002) are missing at the Piuspuszta section, suggesting a more distal and fully marine position. The mollusk assemblage from the sediment surrounding the bioherms is dominated by the mudwhelk *Potamides disjunctus*. This species is eponymous for the *Potamides disjunctus* assemblage defined by Harzhauser and Kowalke (2002), which occurred in littoral to shallow sublittoral settings of the Paratethys Sea on sand under marine conditions. Normal marine and even hypersaline conditions were documented by stable isotope data for this assemblage, based on material from the 'Altes Zollhaus' section

(Latal et al., 2004). The also enormously abundant mudwhelk *Tiaracerithium pictum* dwelled on the surrounding mudflats (= *Granulolabium bicinctum* in Harzhauser and Kowalke 2002 and Latal et al. 2004). Thus, normal marine or even hypersaline coastal marine conditions can be assumed for the phase of bioherm development. These were bracketed by small but rapid oscillations of the relative sea level as indicated by root horizons and erosive boundaries. The frequent caliche layers point to high evaporation due to overall arid conditions during the latest Sarmatian (Piller and Harzhauser, 2005; Koleva-Rekalova et

al., 2021). The low amount of siliciclastic material within the peloidal limestones suggest little terrigenous input and oligotrophic conditions.

## 3 Material and methods

Four bioherms have been detected during fieldwork in 2004 in close distance to each other (bioherms 1–4 in Fig. 3). Two of these bioherms were cut more or less centrally, exposing full vertical sections in the field (bioherms 1 and 2 in Figs 3a–d). The

other two bioherms were recorded as marginal sections (Figs 3e–g). Later stages of excavations documented a comparable size and morphology of these bioherms as for bioherms 1 and 2. The bioherms have been analysed and measured during field work and sediment bulk samples have been taking from each unit as defined in Fig. 2. One of the bioherms (1 in Fig. 3, a–b in Fig. 4) was excavated for further analysis in the laboratory. In a first step, the vertical section was smoothed mechanically in the field, and the surface was impregnated with acrylic varnish with an air compressor. After drying of the varnish, the area

was covered by gauze bandages, impregnated again with varnish, and fixed to a wooden plate. Finally, an about 10–20 cm thick layer of the wall was undercut, tilted with the wooden plate, and transported to the Natural History Museum Vienna (NHMW) where it was prepared in the laboratory. This slab is now stored in the Geological-Paleontological Department of the NHMW. All illustrated thin sections (5 × 5 cm) have been made from samples taken during preparation of this slab. Overview pictures of thin-sections were made with a Zeiss Discovery V20. Thin-section studies were carried out with a Leica

MZ16 and a Zeiss Axioplan2 microscope using the ImageAccess ver. 5 rel. 186 for measurements as well as with a digital microscope Keyence VHX-6000 with included software for photographic documentation and measurements. The amount of vugs in the thin-sections was evaluated with ImageJ version 1.53u. Microbialite terminology follows Aitken (1967), Kennard



and James (1986), Burne and Moore (1987), Riding (2011), Shapiro (2000), Grey and Awramik (2020), Bourillot et al. (2020) and Vescogni et al. (2022).

## 4 Results

The most striking macroscopic constituents of the bioherms are small serpulid tubes dextrally coiled (mostly below 1 mm in diameter), relatively massive shells with prominent crests and prominent transverse rugae (Fig. 5). The imprints of the lower side document that the tubes have been attached to hard or, at least, consolidated substratum. The characteristic morphology allows an identification as *Janua heliciformis* (Eichwald, 1830) (cf. Radwanska, 1994) (Fig. 5), which is a widespread and gregarious species in Sarmatian strata of the Paratethys Sea (Boda, 1959; Piller and Harzhauser, 2005).

### 4.1 Bioherm morphology

The bioherms are characterized in vertical section by their outline, which is reminiscent of stacked bowls. The structures attain about 45 cm in diameter and about 40 cm in height (Fig. 3, 4). The morphologies suggest four growth stages:

1.      The bioherm growth commenced abruptly above a minor erosional surface and is characterized by an open porous fabric with abundant *Janua* shells and a large amount of vugs, which are very vaguely horizontally oriented without forming lamination (Fig. 6). During this initial stage, the bioconstructions attained about 15–20 cm width (Fig. 4).

2.      Subsequently, the bioherms expanded and graded into structures of about 30 cm diameter lacking any macroscopic internal structures. The rapid increase in diameter results in a depressed vase-like outline in vertical section (Fig. 4).

3.      During the next growth stage, the central part of the microbialites continued to grow and widened in diameter. In addition, 'branches' developed along the periphery in vertical view. In horizontal view, these branches reflect a continuous ring-like rim of about 5–8 cm width, separated from the central part of the bioherm by sediment. Two phases of ring-formation can be observed in bioherm 1 (Fig. 4b and three phases in bioherm 2 (Fig. 4d).

4.      Finally, the central part of the thrombolites was successively shrinking whilst the peripheral ring increased in diameter. In the terminal phase of growth, only small patches of the ring and/or of the central part persisted, until they finally became completely covered by sediment (Figs 4e–f).

### 4.2 Bioherm composition and embedding sediment

The wall of the bowl-shaped bioconstructions is built by spirorbid tubes and peloidal, microclotted sediment embedded in clear to bright yellowish "matrix" with a high amount of irregularly shaped vugs with irregular outlines (Fig. 7). No filaments or filamentous structures were detected. The microclots (sensu Frasier and Corsetti, 2003; Harwood and Sumner, 2012; not mesoclots sensu Grey and Awramik, 2020 because of its smaller size) range mainly from 40–120 µm and have a fuzzy outline (Figs 7–9). They are surrounded by acicular (fibrous) cement (Figs 7d, 9) and form also microspherolites around the clots (Fig. 9) what is the reason for the light appearance in thin-section. The interior of the spirorbid tubes is generally free of microclots



and the cavities within the tubes are, partly, filled by acicular cement (Figs 7d, 8). These cements were probably in both cases originally aragonite. The vugs are lined by sparitic dog-tooth cement (Fig. 7d) and lack acicular cement. Serpulid tubes outside

the bowl-shaped structure also lack acicular cement and are lined by dog-tooth cement (Figs 7d, 8d). The boundary between the bioconstruction wall and the underlying peloidal, mesoclotted sediment is sharp but irregular, the contrast between acicular and dog-tooth cement is also very pronounced (Figs 7a–c). Between "bowl wall" and underlying sediment usually a now cement-filled narrow gap does exist (Figs 7b–d) and in some cases an erosional contact may occur (Fig. 6). The vugs are usually randomly arranged only the initial basal part of the bioherm a vague horizontal orientation of large vugs occur (Figs 6,

7a). Vugs account for up to 21% in the early stages of bioherm growth. Terminologically the described structures are somewhat difficult: they resemble a thrombolite but following Shapiro (2000) a thrombolite is a "microbialite composed of a clotted mesostructure (mesoclots)" (Shapiro, 2000, p. 169). The described spirorbid microbialite, however, does not consist of mesoclots but show a microstructure made of microclots, serpulid tubes and acicular cements (Fig. 8). Microclots are rarely arranged in mesoclots but coincide with the microstructure depicted by Shapiro (2000, Fig. 2).

In the surrounding area of the bowl-shaped structure but also within the bowl the sediment is peloidal but generally coarser grained (mesoclots) as in the structure of the 'bowl wall' and contains abundant ostracods, foraminifera (miliolids, rotalids) (Figs 7a, d), a few molluscs and fish teeth/otoliths as well as coarser grained, rounded clasts of microbial origin (thrombolitic/stromatolitic) (Fig. 6). Polymictic terrigenous components also occur. The mesoclots reach 2 mm in size sometimes several mm in elongated shapes and are frequently fused into polylobate grains resembling grapestones (Figs 7b,

10). In contrast to the substance of the bowl-shaped body the cements between the surrounding sediments are coarse grained, dog-tooth cements (Fig. 10). The larger components frequently show oolitic or microbial coating (Figs 7f, 10c, d). The cement in the polylobate components seems to be granular but acicular shapes cannot be fully excluded. Overall, the sediment is a coarse grain- to floatstone (Figs 6, 10) and vug area ranges from 10 to 17%. Due to these characteristics the sediment within the 'bowls' and in the surrounding areas can be classified as thrombolite.

In summary, the herein described microbialites have to be differentiated into the bowl-shaped structures which are dominated by spirorbid tubes, microclots and acicular cements and a typical thrombolite composed by mesoclots which reflects the surrounding sediment and the sediment within the 'bowls'. The Janua-microbialites can be assigned either to "Metazoan-bearing Microbial Structures" or to "Microbial-bearing Metozoan Structures" after classification of Kennard and James (1986).

## 5 Discussion

The spirorbid *Janua* is a filter-feeding, cosmopolitan serpulid genus with wide ecological range, occurring from intertidal to deep sublittoral zones (Knight-Jones et al., 1975; Knight-Jones and Knight-Jones, 1977). The genus is not selective concerning its substrate, but settlement is bound to the presence of bacterial films (Kirchman et al., 1981). Thus, the ecological requirements of extant *Janua* species are not very indicative for paleoecological interpretations. As stated by Ten Hove and van den Hurk (1993, p. 31) "Recent spirorbids never are 'reef-forming'. Nevertheless, records of 'spirorbids' as contributors

to fossil 'spirorbid'-algal stromatolites, forming monospecific banks or even 'reefs', are numerous" ranging from the Devonian





to the Miocene. Under dispute are, however, their salinity requirements, in particular in the Paleozoic, but "Recent spirorbids are marine, or at most brackish species." (Ten Hove and van den Hurk, 1993, p. 31). In the herein studied bioherms foraminifers, potamidid gastropods and cardiid bivalves occur in subordinate numbers within the bioherm-carbonate but are very frequent in the surrounding sediment pointing to full marine or even hypersaline conditions (e.g., Harzhauser and
Kowalke, 2002; Latal et al. 2004; Piller and Harzhauser, 2005).

## 5.1 Growth dynamics and environmental indications of the bioherms

All bioherms started to grow close above the boundary between units 3 and 4. Unit 3 is formed by weakly solidified peloidal carbonate sand (Fig. 4a) and none of the bioherms is in direct contact with this sediment, although the structures are sunken into unit 3. The boundary between unit 3 and 4 is an erosional boundary and caliche formation can be detected laterally. Unit
3 is abruptly overlain by up to 10 cm of a fining upward layer of grain supported fine gravel and peloidal sand with cross bedding (unit 4a). The topmost part of this unit consists of fine peloidal sand and silt with miliolid foraminifers, mollusks and scattered serpulids. The first microbial film developed on this fine sediment (Figs 4b, 6). After subaerial exposure of unit 3, indicated by the erosional surface and the laterally occurring caliche, growth of the bioherms started after reflooding what can be considered a 'start-up' phase. This terminology of Neumann and Macintyre (1985) and Macintyre and Neumann (2011)
was introduced for coral reefs indicating their development in respect to rising sea-level. In our example, however, the bioherm development is mostly depending on sediment production rate. This is documented by the sediment onlapping on the bioherm during this early stage, indicating that the bioherm was keeping up with sedimentation without protruding much above the sea bottom. Thus, the living part of the bioherms was only slightly elevated over the surrounding sediment surface. Onlapping sand at the terminal part of central body in bioherm 1 indicates a relief of less than 5 cm (Fig. 4 inset). Nevertheless, in reference
to the coral reef-classification of Neumann and Macintyre (1985) and Macintyre and Neumann (2011), we identify this growth stage as 'keep-up' stage because the bioherm was still able to keep up with the pace of sediment production (Fig. 11). During this 'keep-up' stage, the bioherms became too heavy to float on the still soft sediment and the structures started to sink. Strong post-depositional deformation of units 3, 4a and of the basal parts of unit 4b documents this process. The uniform growth of coherent, ellipsoidal bioherms changed into the bowl-shaped stage which may have shown some similarities to microatolls or
'thrombolitic microatolls' following the terminology of Burne and Moore (1993) (Fig. 11). The bioherms struggled to keep up with the suddenly increasing subsidence/sediment production and parts of the central area became buried. Subsequently, the process of sinking ceased, as documented by non-deformed bedding surrounding the bioherms. Unlike coral microatolls the top of the bioherm never reached the sea-surface and projected only a few centimeters above sea floor but grew lateral and upwards forming the typical rim of microatolls and dead central areas which became buried (e.g., Melzner and Woodroff,
2015). In convergence with the reef evolution nomenclature of Neumann and Macintyre (1985) and Macintyre and Neumann (2011), we call this phase 'give-up' stage (Fig. 11). In terms of coral microatolls this terminal phase could be morphologically classified as 'Upgrown' microatolls which have low centres encircled by higher living rims, indicating a rising sea level (Woodroffe and Webster, 2014; Melzner and Woodroff, 2015) but sea level may be of minor importance compared to sediment



accumulation in the Sarmatian microbial examples. In contrast to coral microatolls and also to microbial microatolls no
concentric rings could be observed or interpreted in the studied material and also the stacked bowl-shape structures are not
present which, in fact, resemble budding as it is well known from invertebrates such as sponges and corals. The stacked pattern
of two documented bowl shaped bioherms showing 3 growth intervals may be interpreted in two ways: the second bioherm
started growing in the centre of the underlying older bioherm while this was still growing at the rim or the second bioherm
started to grow when the first already stopped growth. The first case indicates a more or less continuous relative sea level
rise/sedimentation rate with the first bioherm already in give-up stage and the second in start-up stage. In both cases growth
sequence could be caused by small-scale fluctuations of the relative sea level but also changing sediment production. Finally,
during the 'give-up' stage, the sedimentation rate exceeded the capability of the bioherm to keep up, leading to the 'escape'
growth structures of the terminal phase of bioherm growth (Fig. 11).

The composition of the microbialites exhibit two main fabric types: *Janua*-microclots-acicular cement boundstone and
thromboidal (mesoclots) boundstone. The first constitutes the bowl-shaped bioconstruction, the later represents the
surrounding sediment and the 'bowl' infill. The *Janua*-microclots-acicular cement boundstone can be clearly related to a
euhaline to hypersaline environment indicated by the mass occurrences of *Janua* tubes and the acicular cements (see also
below). The acicular cements surround the microclots producing microspherulites and solidify the *Janua*-microclots
framework. Since the nearly planispiral tubes of *Janua* produced imprints in the boundstone its (partly) solidification is
documented. The sediment outside the 'bowl' is a thrombolite constructed by mesoclots which are loosely bound by micritic
cement forming not only single peloids but also polylobate, grapestone-type particles. Within and besides the mesoclots
foraminifera, molluscs and also ooids occur demonstrating the marine environment. The particles are also connected with
micritic meniscus-like cements frequently forming bridges over larger distances. This sediment may have formed a cohesive
microbial mat stabilizing the substrate and grew rapidly upwards penecontemporanously with the 'bowl-structure'. The
interparticle spaces but also space within the *Janua* tubes and vugs are lined by dog-tooth cements. These cements indicate
formation in a phreatic environment, mostly marine- but also meteoric-phreatic (e.g., Flügel, 2010; Andrieu et al., 2017).

## 5.2 Serpulid-microbialites in the Paratethys Sea

Microbialites with serpulids and bryozoans have been frequently described from the lower Sarmatian of the Central Paratethys
(e.g., Pikija et al., 1989; Bucur et al., 1992; Friebe, 1994a, b; Harzhauser and Piller, 2004 a, b; Piller and Harzhauser, 2005;
Cornée et al., 2009; see Piller and Harzhauser, 2023 for review) and from the Volhynian and Bessarabian (Middle and Upper
Miocene) of the Eastern Paratethys (e.g., Andrussow, 1902–1909; Andrusov, 1936; Pisera, 1996; Saint Martin and Pestrea,
1999; Daoud et al., 2006; Jasionowski, 2006; Taylor et al., 2006; Studencka and Jasionowski, 2011; Jasionowski et al., 2012;
Górka et al., 2012; Goncharova and Rostovtseva, 2009). But all these bioherms differ considerably from the herein discussed
bioherms by the dominance of bryozoans, such as *Cryptosula* and *Schizoporella* and near absence of microclots and acicular
cement. In addition, the serpulids are represented by *Hydroides*, a serpuline, instead of *Janua*, a spirorbine. Modern
counterparts of such bioconstructions have been termed bryostromatolites by Harrison et al. (2021) and Piller and Harzhauser





(2023) applied this term for the widespread lower Sarmatian bioherms. The driving force triggering the development of these *Hydroides*-bryostromatolites was considerable eutrophication (Piller and Harzhauser, 2023). In addition, polyhaline conditions prevailed, to which *Hydroides* and the bryozoans have been adapted (Piller and Harzhauser, 2023). These environmental

conditions are in strong contrast to the oligotrophic and fully marine to hypersaline conditions in which the *Janua*-microbialites had formed. Comparable structures are rarely reported from Paratethyan deposits. Possible occurrences are those by Hoernes (1898, p. 60) and Papp (1956, p. 58 f.) who described "Spirorbiskalke" (*Spirorbis* limestone) represented by elliptical structures up to 40 cm with dominating spirorbid tubes in an (unspecified) algal limestone. Koleva-Rekalova et al. (2021) documented several elliptical to circular serpulid bioherms from the Bessarabian of NE Bulgaria of up 40 cm diameter. Based on the

illustrated thin-sections, the serpulids in these bioherms are rich in *Janua* and their tubes are filled by fibrous cements (not sediments as mentioned in Koleva-Rekalova et al., 2021) what matches the herein described *Janua* microbialites. Interestingly, these bioherms are also sunken into the underlaying sediment due to their weight (Koleva-Rekalova et al. 2021, fig. 1a).

The reason for the scarce fossil record of upper Sarmatian microbialites is that only comparatively few uppermost Sarmatian deposits of the *Sarmatimactra* Zone are preserved in the Central Paratethys, due to considerable erosion at the

Sarmatian/Pannonian boundary (Harzhauser et al., 2020). The lower Sarmatian deposits with *Hydroides*-bryostromatolites, in contrast, are widespread and well preserved.

## 5.3 Modern and fossil analogues

Typical morphologies of extant microbialites range from domal, discoidal and tabular to ridges and extensive platforms (Siqueiros-Beltrones, 2008; Jahnert and Collins, 2012; Suosaari et al., 2016; Louyakis et al., 2017; Paul et al., 2018, 2021).

Numerous Neogene to Recent thrombolite-bearing deposits have been described in the literature.

The classical modern microbialite occurrences in the Bahamas and in Shark Bay, Australia, contain well studied stromatolites and thrombolites (e.g., Dill et al., 1986; Reid et al., 1995 2000; Andres and Reid, 2006; Planavsky and Ginsburg, 2009; Jahnert and Collins, 2012; Suosaari et al., 2016). Microbialites in Hamelin Pool in Shark Bay show variation in morphologies clearly related to their occurrence from the upper intertidal to the shallow subtidal (Jahnert and Collins, 2012). The deepest subtidal

thrombolites occur down to 6 m water depth and are represented by "non-laminated cryptomicrobial (Cerebroid) structures" and "Microbial Pavement … with tabular and blocky surface morphologies." (Jahnert and Collins, 2012, p. 127). The cerebroid types contain abundant serpulid skeletons and fibrous aragonite cement (Jahnert and Collins, 2012, Figs. 13D, 15K, L). Suosaari et al. (2016) also deal with microbialites in Hamelin Pool and focus on microbial mat communities but also on pervasively precipitated microbial micrite which are all built of aragonite (p. 8), but also acicular aragonite cements occur (Fig.

7f). Although the results of the various authors do not fully coincide, the subtidal microbialites (Cerebroids) contain abundant serpulids and acicular aragonite cement (Jahnert and Collins, 2012) and clearly show a high amount of precipitated micrite (e.g., as clots) and also aragonitic cements (Suosaari et al., 2016). The microbialites in Shark Bay are no direct analogues of the Sarmatian microbialites but the serpulid-rich structures in shallow subtidal settings, abundant micrite precipitation and aragonite cement are features which show strong similarities. The best known subtidal microbialites from the Adderly tidal





channel at the Great Bahama Bank show also very similar internal structures with a thrombolitic fabric of clots and micritic and fibrous aragonite cements both at the microscale and mesoscale (Planavsky and Ginsburg, 2009, Fig. 10, 12). Similar structures are also found in the intertidal microbialites at Highborne Cay, Bahamas (Reid et al., 1999, 2000; Myshrall et al., 2010).

A modern example containing well developed thrombolites is reported from the southern shore of the Persian/Arabian Gulf
close to Abu Dhabi, United Arab Emirates (Paul et al., 2018, 2021). The thrombolites occur in a sabkha setting with water salinities of 75–93 psu and morphologies include domes (5–20 cm in width and a relief of 10–25 cm above a hardground) and bands. Internally, the domes represent stromatolites at the base overlain by thrombolites. Grains are coated by an isopachous acicular aragonite cement with fibrous needles oriented perpendicular to the grain surface (Paul et al., 2018, 2021). The skeletal assemblage is dominated by foraminifers and ostracods and contain subordinate bivalves. Although the growth morphology
and the intertidal setting is not comparable to the Sarmatian examples described herein, the microclots and acicular cement (Paul et al., 2021, Fig. 12 B–D) are strikingly similar and it is a hypersaline environment.

Microbialitic bioherms described from Mediterranean lagoons of southern France are no analogues for the herein described microbialites and also not for other Sarmatian microbialites because Sarmatian bioherms show "mainly clotted and/or peloidal micrite … and, also, fibrous calcite rims around serpulid tubes" (Saint Martin and Saint Martin, 2015, p. 67).
Interesting examples of metazoan/microbial bioconstructions are well documented from Mediterranean caves (e.g., Guido et al., 2017, 2022; Rosso et al., 2021). These are known as biostalactites and are composed of a variety of metazoans besides microbialites. In some examples, the core of the biostalactites is built by a serpulid boundstone dominated by *Protula* tubes but also microbialite/skeletal boundstones and pure microbialite boundstones occur. These structures are comparable with the Sarmatian thrombolites in their microstructure of clotted peloidal limestones and fibrous cements, however, the metazoan
diversity is comparably high and even serpulids are diverse. In addition, the environmental conditions are completely different because these bioherms form all in caves and the microbialites are built by heterotrophic bacteria (Guido et al., 2022).

Two well studied modern examples of microbial microatolls will be cited here: Lake Clifton, Western Australia and Laguna Bacalar, Mexico. In Lake Clifton thrombolitic microbialites showing tabular or concentric, ring-shaped growth of up to 50 cm high developed across a shallow platform which is called reef platform due to the occurrence of the microbial microatolls
(Burne and Moore, 1993). The morphology of the thrombolites varies along the platform depending on proximal or distal location. Lake Clifton is a hypo- to hypersaline lake and the microatolls occur in water depth mostly less 1 m (Burne and Moore, 1993). The water level fluctuates approximately 1 m seasonally and salinity ranged between 15 and 40 psu (Moore and Burne, 1994), however, summer salinity changed between 1983 to 1999 from 29 psu to 48 psu (Konishi et al., 2001). Although metazoans are reported from the thrombolites none of them were dominant (Moore and Burne, 1994; Konishi et al.,
2001). Cementation happens by aragonite crystals. Despite some morphologic similarities between the Sarmatian *Janua*-microbialite and the Lake Clifton thrombolites, the driving processes seem to be different. Although the *Janua*-thrombolites developed in a very shallow marine environment, no sedimentological features indicate that the bioherms developed in the intertidal zone being limited in growth by the low tide level. The central biostrome areas became abandoned due to



sedimentation load and not due to emergence. The Bacalar Lagoon, Yucatan Peninsula, Mexico, harbours the largest known
Holocene freshwater microbialite system (Gischler et al., 2008, 2011; Yanez-Montalvo et al., 2020). The thrombolitic
microbialites include domes, ledges and oncolites, the domes reach a size up to 3 m in height and diameter and they occur
down to 3 m water depth. Maximum depth of the lagoon reaches 15 m, and the water level is fairly constant with annual
fluctuations of 30 cm (Gischler et al., 2011). In the southern part of the lagoon large microatolls (called mini-atolls by Gischler
et al., 2011) are present reflecting a variety of morphologies with concentric flat tops but also with deep central depressions
and cone shaped structures. Mineralogically the microbialite cements are made of calcite (Gischler et al., 2008). Both locations
(Lake Clifton, Lagoon Bacalar) exhibit similarities in growth morphology containing microatolls, in Lake Clifton marine-
derived saline lake water shows elevated calcium carbonate contents (Moore and Burne, 1993, 1994), in Lagoon Bacalar the
microbialites occur in freshwater but also show an elevated carbonate content (Gischler et al., 2008, 2011) is and also the biotic
composition is different. Compared to the Sarmatian bioconstructions no major content of serpulids or other dominant
metazoans is present.

Another distantly similar example are Holocene serpulid-stromatolites from SE Tunisia. These developed in a lagoonal setting
with high salinity and high evaporation and were associated with large populations of the mudwhelk *Pirenella conica* (Davaud
et al., 1994). Like in Piuspuszta, the bioherms are associated with miliolid and rotalid foraminifers and '*Spirorbis*-type'
serpulids. These structures differ, however, in the sequence of growth in which stromatolites overgrow serpulid bioherms but
the microbialites are very similar in being composed of clotted fabric and fan-shaped and aragonitic microspherulites (Davaud
et al., 1994, Fig. 9).

The famous serpulid-tufa bioherms in the Dominican Republic described by Winsor et al. (2012) are only superficially
reminiscent of the Sarmatian bioherms and differ considerably in their meter-scale dimension and various shapes which are
not represented in our examples. The petrographic description is not detailed enough for a direct comparison and also
terminology used is not comparable to other examples. The bioherms formed during the Holocene in restricted hyposaline
lakes along the Enriquillo Seaway (Winsor et al., 2012).

Microbialites with a wide range of morphologies and dimensions with various fabric types partly with a high contribution of
serpulids occur in Messinian oolite shoals in southeastern Spain (Feldmann and McKenzie, 1997; Lipinski, 2009; Goldstein et
al., 2013; Bourillot et al., 2020). Like the Sarmatian bioherms, these bioconstructions formed in a shallow water environment
in less than 10 m water depth but below the intertidal zone under normal marine to hypersaline conditions (Feldmann and
McKenzie, 1997; Lipinski, 2009; Bourillot et al., 2020). Also, the meso- and microscale fabrics are similar. In contrast to the
Sarmatian thrombolites, these Messinian-Mediterranean bioherms contain also calcareous red algae and corals such as *Porites*
and *Tarbellastrea*. In addition, they attain larger dimensions and are associated with *Porites* patch reefs and stromatolites
forming ecological successions related to salinity fluctuations (Feldmann and McKenzie, 1997). Unsurprisingly similar to the
Spanish examples are upper Messinian microbialites from southern Italy also exhibiting various morphologies, fabrics and
diverse biota, including corals, coralline algae, and serpulids (Vescogni et al., 2022). Thus, the Messinian microbialites are





comparable with the Sarmatian bioherms in terms of their subtidal building environments, in meso- and microfabrics, but differ in size and biotic composition and complexity of depositional environments.

None of the microbialites described in the literature display the stacked-bowl vertical-section of the Sarmatian biostromes from NW Hungary as described herein. Fabric composition as well as microbial and acicular cements are in some cases conformable as are subtidal, normal marine to hypersaline conditions and high alkalinity, but morphology does not fit. Overall and in consistency with Piller and Harzhauser (under review) the entire shallow water area in the late Sarmatian of the Vienna Basin (and eventually in all the Paratethys) was dominated by microbial sediments because also most of the widespread oolites are microbially bound as are the bioconstructions (Piller and Harzhauser, under review).

## 6 Conclusions

In the Central Paratethys, *Janua*-microbialites were restricted to the late Sarmatian, whereas bryostromatolites occurred during the early Sarmatian. This exclusive stratigraphic succession coincided with a change from polyhaline eutrophic to normal marine or hypersaline oligotrophic conditions caused by climatic warming and enhanced aridity.

The macroscopic morphology of the *Janua*-bioherms suggests a succession of four growth stages ranging from a 'start-up' stage with small, vaguely laminated bioherms grading into broad bowl-like morphologies ('keep-up' stage) when the bioherms managed to balance the sedimentation rate. The third stage is characterized by a circular rim and a narrow central protrusion. During this stage parts of the central part of the bioherms became covered by sediment, limiting the area of biologically active zones. Two or three phases forming a collar-like structure from the central body are observed in the Hungarian bioherms, resulting in a characteristic stacked-bowl vertical-section. During the final phase, the thrombolites are struggling to keep up with the sedimentation, resulting in successive covering of the bioherm. During this 'give-up' stage, the bioconstructions form small isolated 'escape' structures and finally become buried by sediment.

The internal composition of the bioherms represents a *Janua*-microclots-acicular cement boundstone whereas the surrounding and 'bowl' infilling sediments are thrombolites consisting of peloids and polylobate particles (mesoclots) cemented synsedimentary by micritic cements. Remaining interparticle space and vugs were later cemented by dog-tooth cements.

Only two occurrences of *Janua*-bioherms are known to us, the one in NW-Hungary described herein and one in NE Bulgaria. In both cases, the bioherms are synsedimentarily sunken in the underlaying sediment, indicating that the bioconstructions started to subside due to their increasing weight. For the herein described *Janua*-bioherms, this sinking seems to have triggered the peculiar bowl-shape, whereas the Bulgarian bioherms could keep up with the relatively lower sedimentation rate and formed domal structures. The bowl-shape growth form of the Hungarian bioherm is a unique feature. The upper Sarmatian bioherms grew in a very shallow subtidal environment and have not been restricted in vertical growth by the low tide level. Instead, the bioherms formed during a minor transgressive pulse in normal marine or hypersaline, oligotrophic waters with elevated alkalinity. *Janua*-bioherms might have been much more common in Sarmatian deposits but might have been confused with the much more widespread and better documented lower Sarmatian bryostromatolites.





Both, the serpulid-microbialites and the surrounding peloidal sediment are expressions of excessive microbial activity.
Similarly, the widespread late Sarmatian oolites are microbially bound. Therefore, we assume that the entire shallow water
area of the Paratethys Sea was dominated by microbial sediments from ~12.0 to 11.6 Ma, coinciding with a phase of warm,
arid climate and oligotrophic waters.

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

**Figure 1:** Position of the Piuspuszta gravel pit in the Eisenstadt-Sopron Basin (modified from Schmid et al. (2001) **(a)** and the 'Altes Zollhaus' section in Austria **(b)**. Detail of the Piuspuszta outcrop with position of the bioherms **(c)** (b and c: © Google Earth, 4/08/2012).



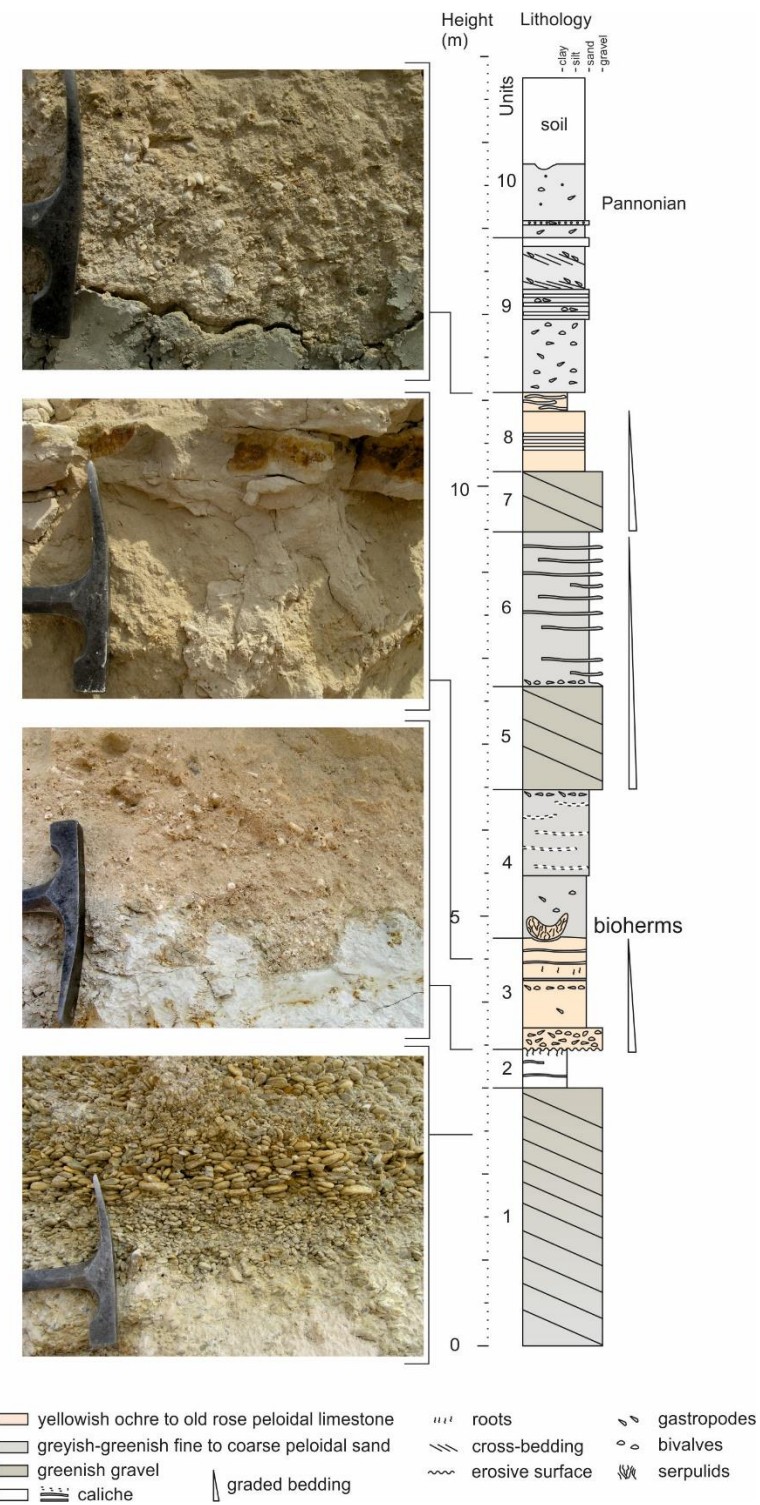

**Figure 2:** The Piuspuszta section based on the outcrop situation in 2004. The bioherms are situated in the base of unit 4.



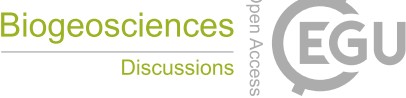

**Figure 3:** Outcrop situation in 2004 and position of the four bioherms (1–4). Bioherms 1 and 2 **(b, c, d)** in the field before bioherm 1 was excavated. Only lateral parts of bioherms 3 and 4 **(e, f, g)** were visible.




**Figure 4:** Bioherms 1 **(a)** and 2 **(c)** and interpretation **(b, d)** of the bioherm and the surrounding sediment. Detail of square in **(a)** showing the onlapping sediment **(e–f)**.



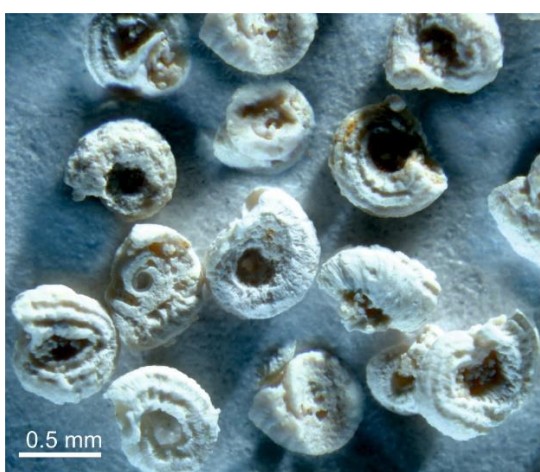

**Figure 5:** Images of *Janua heliciformis* (Eichwald, 1830) collected from the buildups.







**Figure 6:** Composite of petrographic thin sections documenting the fining upward sequence of unit 4a and the initial settlement by microbialites with serpulids **(a).** The same section in dark-field microscopy, documenting the onset of thrombolite growth (blue) versus underlaying sediment (orange) **(b).**







**Figure 7:** Irregular boundary (red dashed line) between *Janua*-microclots-acicular cement boundstone in the upper part and the underlying thrombolite of single and polylobate mesoclots **(a)**. In the *Janua* bioherm the great number of *Janua* tubes is evident and relatively large open vugs are present, which are indistinctly horizontally arranged. In some vugs geopetal infilling of peloidal mesoclots occur (red arrowheads). The thrombolitic sediment at the base is rich in miliolid foraminifers. Transmitted polarized light **(a)**. Detail of the irregular boundary (red dashed line) with thrombolite at the base with single (me) and polylobate (po) mesoclots and miliolid foraminifers (m) and ostracods (o) **(b)**. In the *Janua* bioherm a worm tube is cut near the coiling axis (ja) and microclots (mi) **(b)**. Higher magnification of the boundary interval with microclots (mi) and acicular cement (ac) in the *Janua* bioherm (upper part) and single and polylobate mesoclots (me), micritic cements and dog-tooth cements (dt) **(c)**. In the thrombolite some terrigenous grains are present; transmitted polarized light **(c)**. Higher magnification of figure 7b (red box) showing microclots and acicular cement (ac) in the upper *Janua* bioherm and mesoclots and dog-tooth cement (dt) in the thrombolite; transmitted polarized light, crossed nicols **(d).** Detail of thrombolite in polarized light **(e)** and under crossed nicols **(f)** composed of single and polylobate mesoclots and contain also cross-sections of *Janua* tubes (ja). The mesoclots show micritic meniscus-like cements (red arrows) and the remaining pores are cemented by dog-tooth cement (dt). The internal space of the *Janua* tubes is lined by dog-tooth cement. Few terrigenous grains are also visible. One polylobate grain shows oolitic coating (oo).









**Figure 8:** *Janua*-microclots-acicular cement boundstone: *Janua* tubes in cross-section (partly) filled with acicular cement oriented perpendicular to the tube wall (ac); miliolid foraminifers settling on the outer side of the wall (m) **(a)**. Microclots are very abundant (mi) and also between microclots acicular cement is present. Acicular cement is indicated with (ac); transmitted polarized light **(a)**. Detail with *Janua* tubes which are filled by acicular cement (ac) and well developed microclots (mi) **(b)**. Pores and voids are filled with dog-tooth cement (red arrowhead). One *Janua* tube shows a flat plane of attachment (red arrow). Transmitted polarized light **(b)**. Detail with *Janua* tubes filled

with acicular cement (jac), abundant microclots (mi) also surrounded by acicular cement (ac); transmitted polarized light **(c)**. Detail with *Janua* tube with acicular cement (jac) and microclot patches (mi) with acicular cement in between (ac) **(d)**. The microclot patch is surrounded by a micritic crust (red arrow), pore space is filled by dog-tooth cement (dt), black areas are empty voids; transmitted polarized light, crossed nicols **(d)**. Lower part of the thin section is composed of *Janua* tubes in cross section and microclots (patches) (mi) **(e–f)**. In the upper right part of the thin section an infilling with sediment of mesoclots (me) is visible to show the size differences between micro- and mesoclots.

Black areas are remnants of grinding powder; transmitted polarized light **(e)** and transmitted polarized light, crossed nicols **(f).**







**Figure 9:** Microclots (mi) surrounded by acicular cement (ac) **(a–b)**; pores are partly filled with dog-tooth cement (dt); transmitted polarized light **(a)** and transmitted polarized light, crossed nicols **(b)**. *Janua* tubes filled with acicular cement (jac) and microclots surrounded by acicular cement (ac) **(c–f)**; remaining pores are filled by dog-tooth cement (dt) **(c)**. Transmitted polarized light **(e)** and transmitted polarized light, crossed nicols **(d, f)**.







**Figure 10:** Thrombolite from the surrounding area of the *Janua* bioconstruction. The sediment is composed of peloids (mesoclots) which are connected by micritic cement bridges or meniscus-like structures (red arrows) **(a)**; transmitted polarized light. Detail of thrombolite with single (me) and polylobate (po) mesoclots; the pore space around mesoclots is rimmed by dog-tooth cement (dt); transmitted polarized light,

crossed nicols **(b)**. Thrombolites of single (m) and polylobate (po) mesoclots. Mesoclots are connected by bridges or meniscus-type micritic cement (red arrows) **(c–d)**. Few terrigenous particles (t) are present and one grain shows oolitic coating, mesoclots are rimmed by dog-tooth cement (dt); transmitted polarized light **(c)**; transmitted polarized light, crossed nicols **(d)**. Detail of thrombolite with mostly polylobate mesoclots (po) which are connected by micritic cement bridges (red arrows) **(e–f)**; large peloid at the lower margin shows oolitic coating; pore space is filled by dog-tooth cement (dt). Transmitted polarized light **(e)**; transmitted polarized light, crossed nicols **(f).**



**Figure 11:** Artistic reconstruction of the *Janua*-bioconstruction and idealized growth succession.