# Peer review of "Serpulid-microbialitic bioherms from the upper Sarmatian (Middle Miocene) of the Central Paratethys Sea (NW Hungary) – witnesses of a microbial sea"

_Biogeosciences, 2023_

## Author Response (AR1)

Dear colleague De Jonge, dear Cindy,

Thanks a lot for the efficient editorial work! I think we have followed the suggestions by you and the reviewers as much as possible. Concerning the 'oligotrophy'-problem, we have now removed the term at places or added 'probably' where appropriate. Generally, such a carbonate factory would not develop in non-oligotrophic settings, but one will always find weird exceptions, which we don't want do discuss herein. It's not a central message of the paper and therefore, we weakened any emphasis on the topic.

I have performed the color-blindness-simulator-test and think that the illustrations will work. For the cathodoluminescence pictures in fig 9, it will be impossible to replace the colors.

Best

Mathias

Vienna, 16.10.2023

Univ. Prof. Dr. Mathias Harzhauser
Head of Department
Geological-Paleontological Department
Natural History Museum Vienna
Burgring 7 – 1010 Vienna – Austria
http://www.nhm-wien.ac.at/mathias_harzhauser
mathias.harzhauser@nhm-wien.ac.at
tel: 0043-1-52177-250

**REVIEW1**

We are grateful to Reviewer 1 for his constructive comments. In the following, we give a point-by-point answer to his suggestions.

The introduction on the paleoecological evolution and bioconstructions of the Central Paratethys Sea is well addressed and introduces the readers to the topic, but the paragraph lacks the aim of the present research, for example:

   why these biconstructions were studied?

   what new aspects want to be evaluated in this research compared to those described in previous papers? [...]

All these aspects were addressed in the following paragraphs but they were not introduced as aims of the research in the introduction.

This is true and therefore we added following paragraph in the introduction:

The paleoecological interpretation of the Sarmatian sea was based so far especially on its mollusc assemblage and foraminiferal faunas (e.g., Harzhauser & Piller, 2007; Kranner et al. 2021 and references therein). The herein observed bioherms, in contrast, have not been described so far. Consequently, no information on the paleoenvironmental indications of these structures is available.  Herein, we want to elucidate the main

constituents of the bioherms to describe composition, structure and mode of growth of these enigmatic bioherms. These informations will give new insights in the marine ecosystems of the Sarmatian Sea.

In the paragraph "4.1 Bioherm morphology" were described the four stage of growth of the bioherm but this is an interpretation. Here it would be more appropriate to describe the morphological and compositional characteristics at the mesoscale of the bioherms. The interpretation must be moved after the result paragraph.

We have removed this part to the Discussion

In general, the microfacies are well described and illustrated but the microbial or organic induced nature of the micrite is based only on morphological observations at mesoscale and thin section observations, so the interpretation as microbialite seems speculative. To confirm their model, the authors should supplement the data with SEM/EDS, Raman spectroscopy, UV-fluorescence or other techniques aimed to put in evidence: 1) details at higher magnification of the micro and nano-structures of the putative microbialite fraction; 2) the mineral composition of the presumed autochthonous micrite; 3) the organic nature of the microbialites.

In addition, following requested information we conducted cathodoluminescence microscopy, however, the material is strongly diagenetically altered and only the youngest (dog tooth) cement generation was clearly discriminable all other components are composed of coarse grained neomorphic spar with only detectable ghost structures of the original crystals (e.g., acicular cement). We added new Fig. 9 to show this textures.

The discussion section is well structured and the comparison between the studied metazoan/microbial bioconstructions and those reported in literature, which spam in time and space, is well addressed. The only doubt is about the support of the discussion with the acquired data, mainly concerning the interpretation of the peloidal and clotted peloidal micrite as microbially mediated. The morphologies seem attributable to microbial activities but more micro- nano-morphological, mineralogical and biogeochemical investigations could prove definitively their origin.

As mentioned above, the original crystals are widely altered by neomorphic processes what does not allow designating microbial carbonate by crystal morphology. The geochemical composition of the sediment is also altered by diagenesis and, in addition, by infiltration of resin into the weakly cemented sediment obscuring the original composition. Geochemical analyses will therefore not deliver reliable results.

**REVIEW2**

We thank Reviewer 2 for his positive feedback and the constructive comments, which we all accepted.

Improvements are possible with respect to the comparison with similar Miocene to present-day microbialites. Although this discussion chapter is well elaborated, it remained without broader implications (except that there are always some minor differences to the investigated serpulid-microbialites from Piuspuszta). In my view, a short discussion on the significance of these (and similar) microbialites in the Phanerozoic record would be desirable, in particular, why calcified cyanobacteria do not occur in the shallow-water serpulid-microbialites described, although microbial precipitates and acicular aragonites are formed.

The lack of filamentous microbial structures may be due to the subtidal environment where the structures formed. Similar to the examples from Shark Bay these subtidal structures may have been dominated by coccoid cyanobacteria as described by Suosaari et al. (2016). We added a sentence pointing at this possibility.

In this context, it would also have been good to measure stable carbon and oxygen isotopes of material of the studied section. Currently, the paper relies on data on gastropods from the adjacent section "Altes Zollhaus", with no data from the microbialites themselves. In my view, it would have been worth trying to obtain separate phases for this purpose. Also a series of mixed samples from the thrombolites would have been quite informative.

This is surely correct but beyond the scope of this current project.

Title: What is the meaning of "microbial sea"? I am not sure if this wording is appropriate: The Sarmatian sea was certainly in the same way populated by microbial communities as present day or other fossil marine settings. Please consider rewording.

We kindly disagree here, because we think that this sea was exceptionally dominated by microbial communities, as shown by the widespread oolites and peloids.

Introduction:

- Please specify here or in the methods chapter to which salinity scale your refer (Venice system? Reference?). This is important especially for the term "hypersaline".

Anonymous: The Venice System for the classification of marine waters according to salinity. Limnology and Oceanography, 3, 3, 346–347. https://doi.org/10.4319/lo.1958.3.3.0346.

- Lines 103 f.: Caliche commonly forms in "semi-arid" rather than in real "arid" environments. What is the indication to infer "arid" conditions in the present case?

We have modified to semi-arid.

- Lines 105 f.: oligotrophic conditions: If I understand correctly, this statement is solely based on the low amount of siliciclastic input? I could imagine that the conditions were indeed rather oligotrophic, but the line of argumentation is not compelling. There are examples of low-siliciclastic peloidal limestones within highly eutrophic settings (e.g. Ries crater lake). Is the absence of organisms indication eutrophication the major argument? Please add a short clarification in the text.

The fact that the siliciclastic input is very low despite the coastal position and the fact that no biota preferring eutrophic settings (as described by us for the early Sarmatian; Piller & Harzhauser 2023) occurred at that time supports our interpretation. In addition, we cite modern analogues from oligotrophic settings (e.g., Siqueiros-Beltrones, 2008; Johnson et al., 2012; Mobberley et al., 2015). As the reviewer himself did not question the oligotrophic setting as such, we prefer to keep the term.

- Line 169 and throughout the text: Why not calling the polylobate grains "aggregate grains"?

We use the term polylobate in accordance with Shapiro (2000) but we also used grapestone in parallel. In addition to this we also added now "aggregate grains" to show the terminological range for such grains.

- Please provide the argument(s) of an increased alkalinity at the time of bioherm development.

Stable isotope data from Upper Sarmatian mollusc shells, collected close to the Piuspuszta section reveal a marked peak in δ13C. which was interpreted by Harzhauser et al. (2007) as indication for elevated alkalinity. In addition, we refer to Pisera (1995, 1996) and Corneé et al. (2009) who also discussed elevated alkalinity for the Sarmatian Sea. This was added in the introduction paragraph.

- Salinity: Since no geochemical/isotope data are available, please expand on the associated fossils. In lines 166 f. you mention "ostracods, foraminifera (miliolids, rotalids), a few molluscs and fish teeth/otoliths". Is there anything known on the identity of the ostracods (which might be very useful for salinity interpretations)? If miliolid foraminifera are abundant, you may simply state that the setting deviated from open marine conditions and was affected by salinity fluctuations. Besides of *Potamides disjunctus*, which mollusc species are present in the bed of the microbialites? Currently I do not see real evidence for "hypersaline" conditions (in the sense of >40 permill salinity). Are there any echinoderm fragments in the thin sections?

We see the point and have changed hypersaline to elevated saline. In addition, we refer to Kranner et al. (2021), who calculated salinities around 37 PSU for late Sarmatian foraminiferal assemblages from the Vienna Basin. Therefore, the term hypersaline might be misguiding.
Ostracods have not been studies in the frame of this project.

Echinoderms became extinct in the Paratethys at the Badenian/Sarmatian boundary. Therefore, they can't be detected at the studied section.

- Terminology: [Not that important...] Please use the term "bioherm" consistently throughout the manuscript. "Bioconstruction" is, as far as I remember various discussions, an ambiguous term as you may consider e.g. metazoan skeletons bioconstruction, too.

We have replaced bioconstruction by bioherm.

- Line 66: replace "Glibert-type" by "Gilbert-type"

done

- Line 85: Please put "Sarmatimactra vitaliana" in italics.

done

- Line 173: In my understanding, these are grain- to rudstones. A floatstone is characterized by large components floating in a fine-grained (micritic or mud) matrix.

changed

- Line 315: Consider replacing "happens by" by "resulted from".

done

- Figure 2: Please increase size and clarity of the symbols for gastropods and bivalves. Consider placing some of the symbols right to the sediment column
(e.g. sedimentary structures, label for caliche, fossils) so that they become more prominent and recognizable to the reader. You may also consider indicating shallowing-upwards sequences and discontinuities where evident. Likewise, the stratigraphic stages Sarmatian, Pannonian, and the corresponding boundary (at basis of unit 10) can be shown right or left to the column. Currently, the term "Pannonian" appears isolated at the level of unit 10.

Done. We followed the suggestions and have simplified the legend a bit.